# Preventing Prejudice Emerging from Misleading News among Adolescents: The Role of Implicit Activation and Regulatory Self-Efficacy in Dealing with Online Misinformation

Giuseppe Corbelli [1,*], Paolo Giovanni Cicirelli [2], Francesca D'Errico [2] and Marinella Paciello [1]

1 Faculty of Psychology, Uninettuno University, 00186 Rome, Italy; marinella.paciello@uninettunouniversity.net
2 Department of Educational Sciences, Psychology, Communication, University of Bari Aldo Moro, 70122 Bari, Italy; paolo.cicirelli@uniba.it (P.G.C.); francesca.derrico@uniba.it (F.D.)
* Correspondence: giuseppe.corbelli@uninettunouniversity.net

**Abstract:** This paper explores the possibility of preventing prejudice among adolescents by promoting the analytical processing of social media content emerging from racial misinformation. Specifically, we propose, at this aim, an intervention that centers on recognizing stereotypical beliefs and other media biases about a group of people in misleading news. To better understand the variables that contribute to improving socio-analytical performance in the face of such misinformation, we investigated the influence of implicit associations as a tendency toward the automatic labeling of groups, as well as two dimensions of perceived self-efficacy in the face of misinformation, one active and one inhibitory. Our results demonstrate the presence of a negative link between affective prejudice and socio-analytical processing, and that this analytical performance toward misleading news is negatively related to the individual tendency toward implicit activation, and is also explained by the inhibitory factor of the perceived efficacy toward misinformation. The role of the active factor related to the perceived ability of fact-checking is not significant. This research suggests that education focused on the socio-analytical processing of misleading news in social media feeds can be an effective means of intervening in online affective prejudice among adolescents; the implications and limitations of our findings for future research in this area are discussed.

**Keywords:** affective prejudice; analytic processing; implicit bias; misinformation; self-efficacy

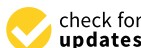



## 1. Introduction

Contemporary research has established a concerning association between the prevalence of misleading news and the emergence of prejudicial conduct and beliefs (Wright and Duong 2021). Consequently, there is a growing interest in exploring interventions to counter the spread of ethnic-based prejudice, particularly among young people who are still developing their social and cognitive skills. Adolescence represents a critical developmental phase wherein individuals are highly susceptible to the formation of prejudicial attitudes and beliefs (Beelmann and Lutterbach 2022). Furthermore, the rise of online prejudice towards outgroups has been identified as a significant issue in social networks (Daniels 2013; Paciello et al. 2021; Ramasubramanian 2007; White et al. 2015), alongside the susceptibility of adolescents to fake and misleading news (Herrero-Diz et al. 2020). A crucial aspect of this investigation involves a better understanding of the mechanisms that shape adolescents' interaction with misleading and false information, especially concerning its impact on the development of prejudiced beliefs. Despite adolescents' recognition of the low credibility of information circulating on social media, they often neglect fact-checking procedures (Papapicco et al. 2022). This discrepancy may be influenced by various factors, such as the desire for social inclusion and the need to share interests with others (Beyens et al. 2016; Notley and Dezuanni 2019). Therefore, it is crucial to investigate the cognitive processes involved in adolescents' engagement with misleading news, particularly in the

context of the new generation of teenagers born in the era of social networks, and who are heavily reliant on them.

In light of an agentic perspective, this study proposes that effective interventions addressing outgroup prejudice in adolescents can be achieved through the promotion of intentional, conscious, and controlled analytical processing. In this way, the adolescent can move beyond the reliance on rapid automatic associations between social groups and attributes, which are often triggered by linguistic and content features characteristic of online misleading news. Instead, the focus is placed on fostering specific self-efficacy beliefs related to an individual's perceived capability in resisting the urge to engage with fake news and actively verifying the sources and contents of such information. By considering these two critical individual factors (see Figure 1) as indicators of distinct modes of information processing, this research seeks to develop tailored strategies aimed at empowering adolescents to effectively self-regulate when confronted with the automatic tendencies commonly elicited by misleading online news. Through this approach, the study aims to mitigate the prejudicial effects that may arise from exposure to misleading information. Thus, by clarifying the role of individual factors and beliefs in the socio-analytical processing of online misinformation, this research contributes to a more comprehensive understanding of the mechanisms underlying the development and potential prevention of prejudice in adolescents.

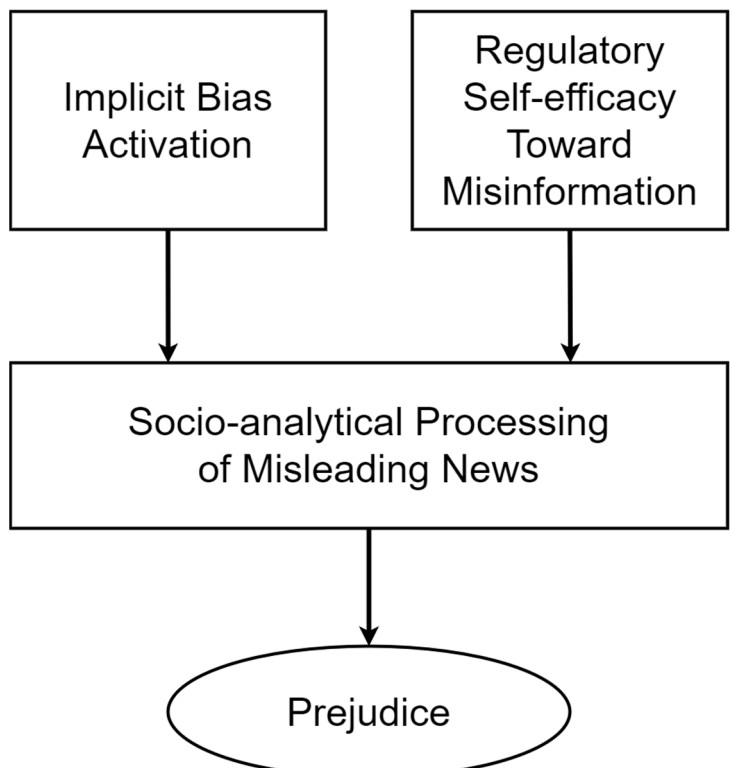

**Figure 1.** Hypothesized model.

### 1.1. Prejudice and Socio-Analytical Processing of Misleading News

Understanding the cognitive processes involved in adolescents' interaction with misleading information on social media is of paramount importance, given the potential reinforcement of biased beliefs and behaviors. Adolescents are increasingly exposed to online sources, and this extensive interaction can subject them to a wide array of information, including misleading news, which may have a significant impact on their socio-cognitive development (Hammad and Alqarni 2021; McGrew et al. 2018; Papapicco et al. 2022). Misleading news, deliberately designed to deceive or manipulate, holds the potential to shape perceptions, attitudes, and prejudices toward various social groups (Pennycook and

Rand 2021). This kind of online misinformation often thrives on automatic and heuristic processing, driven by individuals' tendency to rely on mental shortcuts and cognitive biases to make sense of complex information quickly (Pennycook and Rand 2019). This tendency makes individuals prone to adopting prejudiced attitudes based on rapid associations between social groups and misleading content (Wright and Duong 2021).

Furthermore, research has consistently shown that adolescence also presents a critical window for effectively addressing prejudice, as interventions targeting stereotypical beliefs and associations in adulthood have proven largely ineffective (Hsieh et al. 2022). Nevertheless, the behavior of adolescents is shaped by a multitude of factors, encompassing the influential role of peer groups (Hjerm et al. 2018), the affordances provided by online social networks, and the impulsive tendencies inherent in this specific developmental stage (Raabe and Beelmann 2011; Daniels 2013). Consequently, devising tailored strategies to address the distinctive characteristics of young individuals becomes exceedingly challenging and necessitates comprehensive planning and understanding. In response to this challenge, socio-analytical interventions have emerged as a promising approach (D'Errico et al. 2023; Paul and Elder 2004; Banas et al. 2020; Birtel et al. 2019). These interventions guide adolescents to critically reflect on misleading news by examining its stereotypical language, its sources, and alternative viewpoints of reported events. By encouraging analytical thought processes, such interventions seek to promote a deeper understanding of the potential biases and distortions present in misleading news (Swami et al. 2014). In this study, we seek to answer the following question: is a stronger capacity for the socio-analytical processing of online misinformation associated with lower levels of affective prejudice against the targeted minority, in this case, migrants? By investigating the role of socio-analytical processing in mitigating affective prejudice, the aim is to uncover potential ways to reduce negative intergroup affect. In fact, previous studies have highlighted that affective prejudice, characterized by negative emotional reactions and hostile attitudes toward outgroup members, can lead to harmful social consequences, including intergroup conflicts and discriminatory behaviors (Dovidio et al. 2017; Pettigrew and Tropp 2006).

### 1.2. Dual Information Processing toward Socio-Analytical Thinking

Moreover, our study aims to contribute further insights on two individual cognitive dimensions linked to information processing that influence the socio-analytical elaboration of misleading news. We ground our investigation in the classical distinction between two systems of information processing: one automatic and rapid, and the other controlled and slow (Petty and Cacioppo 1986; Strack and Deutsch 2004). We hypothesize two specific pathways influencing an individual's capacity for the socio-analytical processing of misleading online news.

On one hand, implicit activation, described as the automatic activation of attitudes or stereotypes about a particular group of people without conscious awareness or intentional control (Greenwald and Banaji 1995; Strmic-Pawl 2021), represents an indicator of the activation of the automatic or heuristic pathway (Fazio et al. 1995; Lai et al. 2013; Nosek et al. 2011). Repeated exposure to stereotypical attribution patterns in childhood leads to the automatic activation of these mental shortcuts associating evaluations with social groups, starting from adolescence and remaining essentially stable into adulthood (Greenwald et al. 1998; Williams and Steele 2016; Baron and Banaji 2006; Rutland et al. 2005). These associations can be effortlessly activated when the individual recalls the social group that is the object of the automatic association, often outside of one's awareness (Degner and Calanchini 2020; Gabrielli et al. 2022; Hofmann et al. 2005). The inherent characteristics of this automatic and effortless activation characterize implicit associations as a type of rapid and automatic information processing. Moreover, in the case of online misinformation, these quick associations can be explicitly triggered by distorted content or specific linguistic biases, typically present by definition in misleading news (D'Errico et al. 2022). Considering the automatic pathway of information processing, we seek to answer

the following question: does an increased implicit activation hinder the socio-analytical processing of misinformation?

On the other hand, individuals are not merely passive recipients of automatic tendencies elicited by the environment but can exercise control over their reactions and behaviors (Bandura 1989). Perceived self-efficacy beliefs represent an individual's degree of confidence in their ability to act on the environment or themselves in line with their goals (Bandura 1977), and it is already known how these beliefs influence the outcome of final behavior (Caprara et al. 2011). Regulatory self-efficacy in dealing with misinformation refers to an adolescent's perceived ability to inhibit the need to share potentially false news, even if doing so might bring social benefits, and to take the time to assess its veracity and consistency with other available information, despite cognitive or social costs. Regulatory efficacy against misinformation represents, in this work, the strength with which a person can voluntarily engage in the controlled and serial pathway of misleading news processing, as previous research has already highlighted the crucial role of inhibitory self-efficacy in preventing the re-sharing of misinformation among adolescents (Paciello et al. 2023). Taking into account the controlled pathway of information processing, our research question is therefore the following: does a stronger belief in one's abilities to deal with misinformation promote the socio-cognitive analysis of online misleading news?

## 2. Methods

### 2.1. Participants

To secure an appropriate sample for the study, multiple educational institutions in Italy were contacted to assess their interest in participating in the research. Subsequently, three schools located in southern Italy expressed interest, and their student populations were utilized as a convenience sample. The final sample comprised 176 participants, of which 90 were males. The average age of the participants in the study was 14.0 years old ($SD_{age}$ = 1.1). Considering the characteristics of the adolescent sample, 162 students (92.0%) identified themselves as belonging to the White/Caucasian ethnic group, 2 students (1.1%) made their ethnicity explicit as Latino/Hispanic, only one student (0.6%) identified their ethnicity as Asian, and 11 students (6.3%) preferred not to express their ethnicity. Regarding family characteristics, most of the adolescents in the sample reported a high school diploma as the highest educational qualification earned by their parents (45.4% for both parents), followed by a middle school diploma (31.6% and 27.0% for both father and mother), then a bachelor's degree (13.2% for father and 19.0% for mother), a master's degree/post-graduate degree (6.9% for father and 6.3% for mother), and finally an elementary school diploma (2.9% and 2.3% for father and mother, respectively).

### 2.2. Procedure

This research project was conducted in accordance with the ethical guidelines delineated by the Helsinki Principles of Ethics and the AIP Code of Ethics; the research was approved by the ethics committee of the university to which one of the authors is affiliated, under the reference code ET-22-01. All participants were minors, and informed consent was obtained from their legal guardians before questionnaires were administered. No participants refused to participate or withdrew from the study after informed consent was received.

The study was divided into two stages; during the first stage, administered through computers in the schools' computer lab, adolescents completed an introductory battery of tests and questionnaires to collect basic socio-demographic information and assess affective prejudice, active and inhibitory regulatory self-efficacy, and implicit bias. The survey tool was implemented through the scientific markup software Inquisit v.6.5.2. The duration of this initial survey was approximately 50 min.

In the second stage, after 1 week, the same students were presented with a new two-part quanti-qualitative analytical intervention tool administered through Google Forms. First, the adolescent was led through an imaginary and playful context simulated through

engaging images and textual descriptions; the request was to imagine oneself as a reporter in a newspaper editorial office during the assignment of the reporter of the week badge. Therefore, together with colleagues, the student was required to further the knowledge and confidence placed in certain news stories, checking their veracity, and being guided in recognizing those verbal-communicative features that characterize misleading news. Particular emphasis was placed on understanding and recognizing the role played by stereotypes in such news stories. Then, the request made to the student within the same simulative context was to actively investigate the veracity of the news sources by reading and comparing other news stories related to the same event, but this time from the perpetrator's point of view. Thus, the request was to comprehensively reformulate the same news story by integrating the two different points of view, purging it of verbal biases and the reliance on stereotypical attributions. The duration of the second stage was approximately 50 min.

At the end of the full procedure, an appropriate and detailed debriefing was conducted with all participants.

### 2.3. Measures

#### 2.3.1. Affective Prejudice

Following Dijker (1987), the level of a generic negative affect towards migrants was assessed by directly asking "How much do immigrants trigger this emotion in you?", followed by a list of six negative emotions (Kessler et al. 2010). The congruence between one's negative emotional state elicited by thinking about immigrants and each given item (fear, distrust, uneasiness, insecurity, anger, and shame) was assessed through a 5-point Likert scale, ranging from 1 (not at all) to 5 (very much). To compute the index, the scores of all items were averaged. Cronbach's $\alpha$ for the six items measured at the first time point was 0.84, while for the same items evaluated after 1 week, it was 0.86.

#### 2.3.2. Socio-Analytical Processing of Misinformation

To calculate an indicative score of the overall ability to process a specific piece of misleading news, an already proven procedure was applied (D'Errico et al. 2023), structured on the basis of the guidance of Paul and Elder (2004), and studies conducted on the topic of mediated contact (Banas et al. 2020; Birtel et al. 2019).

Initially, with a guided procedure encompassing a series of tasks with examples and explanations, participants were encouraged to engage in a process of a multidimensional analysis of a racial hoax (Figure 2) administered to them through the descriptive scenario. Each participant worked with a single racial hoax, assigned through randomization. Throughout the study, 10 racial hoaxes were administered, with variations related to the perpetrator of the moral violation or content.

After data collection, a coding scheme was applied to measure the participant's responses to each task. Two independent judges coded the different tasks according to a specific scheme: stereotype recognition in the title (from 0 = no to 1 = yes), with an inter-rater reliability of k = 0.852; stereotype recognition in the text (from 0 = no to 1 = yes), with an inter-rater reliability of k = 0.627; distinguishing between fact and judgment (from 0 = none and 1 = only one to 2 = both), with an inter-rater reliability of k = 0.767; source understanding (from 0 = no and 1 = detected but in a wrong way to 2 = correct), with an inter-rater reliability of k = 0.934; other focus identification (from 0 = no and 1 = incorrect to 2 = correct), with an inter-rater reliability of k = 0.932; and identification of other reasons and reflecting on the existence of alternative focus, with questions such as "Is there an alternative narrative? Is there any clue?" (from 0 = no to 1 = yes), with an inter-rater reliability of k = 0.944.

After answering the previous questions, the participants were shown a counter-story (Figure 3), in which the protagonist of the previous stimulus (i.e., the racial hoax) explained his own standpoint on the same fact, from a reliable journalistic source. The idea underlying this procedure was to humanize the immigrant, identifying him as an individual by

giving him a specific name and story (son, father, and boy scout) based on the description of personal details that would try to give a counter-stereotypical image and emotional experience of him in order to induce empathy. If in the misleading news he was an illegal immigrant, here he is a good person helping others, concerned for the safety of his son.

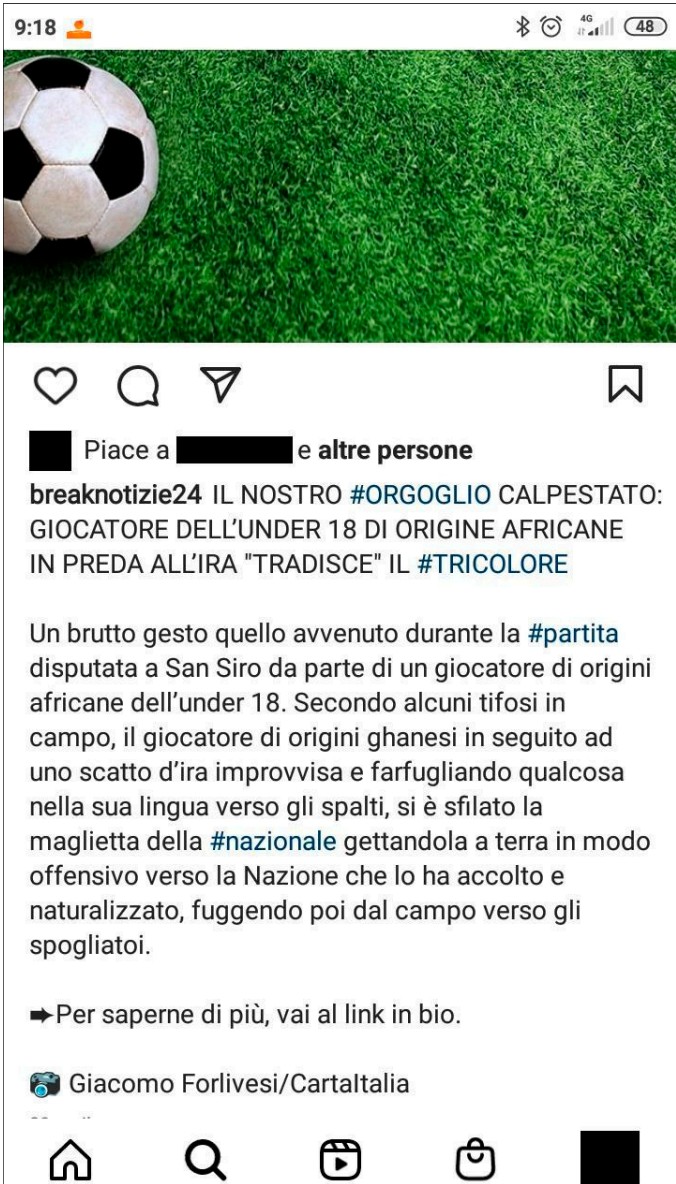

**Figure 2.** Racial hoax example.

Finally, participants were asked to rewrite the story, taking into account all the points discussed and following the reading of the counter-story.

Peculiar aspects of the title and of the text of the rewritten news were therefore evaluated and subsequently codified by two independent judges: rewriting the title focus (from 0 = same focus and same judgment to 4 = multiple foci and no judgment), with an inter-rater reliability of k = 0.743; rewriting the text focus (from 0 = same focus and same judgment to 4 = multiple foci and no judgment), with an inter-rater reliability of k = 0.80; the presence of stereotypes in the rewritten title (0 = negative stereotype, 1 = no stereotypes, and 2 = positive stereotype), with an inter-rater reliability of k = 0.965; the presence of stereotypes in the rewritten text (0 = negative stereotype, 1 = no stereotypes, and 2 = positive stereotype), with an inter-rater reliability of k = 0.98.

Finally, the Socio-Analytical Processing of Misinformation index was obtained by summing the scores obtained from the various tasks, after having standardized them to avoid possible bias effects due to the different measurement scales of the individual items for each task.

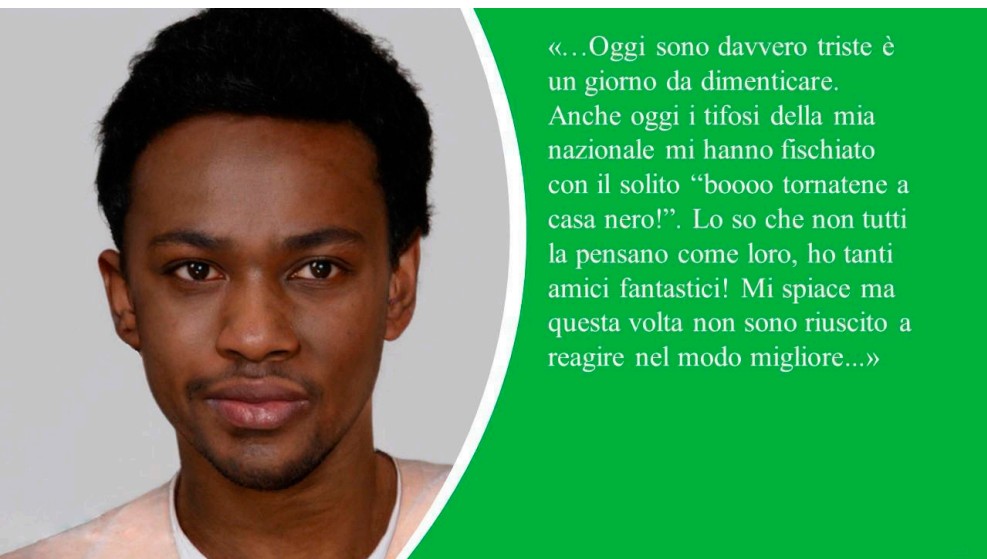

**Figure 3.** Alternative news item based on the immigrant's point of view.

### 2.3.3. Implicit Bias Activation

To assess individual differences in the automatic activation of cognitive associations, the Brief Implicit Association Test, or BIAT (Sriram and Greenwald 2009), was employed. The BIAT is a shorter version of the Implicit Association Test (Greenwald et al. 1998), designed to measure implicit cognitive associations more efficiently. Like the IAT, the BIAT is a reaction-time-based categorization task that assesses the differential associative strength between bipolar targets and evaluative attribute concepts, providing an approach to indexing implicit beliefs or biases without relying on self-disclosure (Healy et al. 2015). First, the D-score was pre-processed as suggested by Greenwald and colleagues (Greenwald et al. 2022). Then, after performing a 5-level ordinal scoring (Sriram and Greenwald 2009), the three intermediate levels indicative of no to moderate bias were subsequently merged with each other. Lastly, the measure's absolute value was calculated to consider the implicit bias activation (hence, IBA) as a measure of the individual tendency to quickly and automatically associate a label with a social group, regardless of the label's valence (i.e., good or bad).

### 2.3.4. Regulatory Self-Efficacy in Sharing Misinformation

The evaluation of adolescents' perceived competence in managing online misinformation was conducted utilizing the 8-item Regulatory Self-Efficacy in Sharing Misinformation scale (Paciello et al. 2023). In accordance with the social cognitive literature (Bandura 1991), this scale encompasses two distinct yet inter-related dimensions: an inhibitory and an active factor. Cronbach's α for the scale was 0.77.

RSSM active factor. The three items of the active factor intend to quantify the perceived ability of teenagers to autonomously examine and verify the truthfulness of news content by checking online sources.

RSSM inhibitory factor. The inhibitory dimension, composed of five items, is designed to gauge the perceived capacity of adolescents to resist sharing a piece of news that seems unreliable, even when it may be advantageous or they experience the impulse to do so.

2.3.5. Data Analysis

Descriptive statistics were initially inspected, followed by an analysis of zero-order correlations. For each variable, skewness and kurtosis values were evaluated, and Mardia's (1970) test was performed to assess the multivariate normality assumption. Subsequently, the mechanism underlying missing data was examined to ascertain if it was missing completely at random (MCAR) or if the pattern of missing data was associated with the variables under consideration (Little 1988). To validate the congruence of the proposed relationships with empirical measures, the hypothesized model was assessed using Structural Equation Modeling techniques (Bollen 1989). To model the stability of affective prejudice over time as the main outcome while controlling for measurement error, a latent variable representing the true underlying construct of interest was built from two indicators of affective prejudice measured likewise at two different time points (Bollen 2007; Muthen and Muthen 2000). The posited model includes the gender, ethnic background, and ethnicity of the perpetrator of the news item as covariates. All the analyses were conducted employing the lavaan package (Rosseel 2012) within the R statistical software environment (R Core Team 2021), along with the mvnormalTest (Zhang et al. 2020), psych (Revelle 2022), and Hmisc (Harrell 2023) packages.

## 3. Results

### 3.1. Preliminary Analyses

Means, standard deviations, and correlations for all the relevant variables for the total sample are presented in Table 1. Specifically, these results show that the two measures of affective prejudice are strongly correlated, as expected; a moderate negative correlation was also found between both prejudice scores and the adolescent's performance at the socio-analytical processing of misinformation. Furthermore, the same socio-analytical processing is significantly (and negatively) correlated with implicit bias activation, while it is positively correlated with both factors of regulatory self-efficacy toward misinformation. Finally, the two different dimensions of this perceived belief are also moderately positively correlated with each other.

**Table 1.** Descriptive statistics and zero-order correlations for the relevant variables.

|  | M | SD | 1 | 2 | 3 | 4 | 5 |
|---|---|---|---|---|---|---|---|
| 1. AP (T1) | 1.28 | 0.27 | – |  |  |  |  |
| 2. AP (T2) | 1.23 | 0.24 | 0.69 *** | – |  |  |  |
| 3. SAPM | 0.11 | 4.84 | −0.34 *** | −0.31 *** | – |  |  |
| 4. IBA | 0.23 | 0.42 | 0.14 | 0.09 | −0.17 * | – |  |
| 5. RSSM-act | 3.23 | 0.77 | −0.07 | −0.03 | 0.17 * | −0.11 | – |
| 6. RSSM-inhib | 3.31 | 0.84 | −0.13 | −0.07 | 0.24 ** | −0.11 | 0.40 *** |

AP: Affective Prejudice; SAPM: Socio-Analytical Processing of Misinformation; IBA: Implicit Bias Activation; RSSM-act: Regulatory Self-Efficacy in Sharing Misinformation (active factor); RSSM-inhib: Regulatory Self-Efficacy in Sharing Misinformation (inhibitory factor). * $p < 0.05$, ** $p < 0.01$, *** $p < 0.001$.

Prior to proceeding with the analysis, the distributional properties of the relevant variables were assessed. While all kurtosis and skewness values ranged below the cut-off of ±1 (Marcoulides and Hershberger 2013; Muthén and Kaplan 1985), Mardia's test confirmed a statistically non-negligible departure from multivariate normality for both skewness (83.6, $p < 0.001$) and kurtosis (2.4, $p = 0.015$). Regarding the analysis of missing data, the findings indicated the existence of MCAR missing data mechanisms operating on the variables examined in the current study, as the result of Little's MCAR test was not significant ($\chi2 = 128.86$, $p = 0.08$). Under this assumption (Marcoulides and Schumacker 2013), the full information maximum likelihood (FIML) estimation of parameters was employed for addressing missing data within the lavaan package (Muthén and Shedden 1999; Schafer and Graham 2002).

### 3.2. Verification of the Theoretical Model

Due to the non-normal multivariate distribution of observed variables, the Structural Equation Model was tested by means of a maximum likelihood estimation with robust (Huber–White) standard errors. Figure 4 shows the diagram with the result of the analysis; for the sake of clarity, covariates are not shown. The proposed model's fit indices were evaluated against the established cut-off values suggested by Kline (Kline 2016), which confirmed the plausibility of the hypothesized relationships between variables: $\chi 2 = 6.409$, df = 9, $p = 0.698$; CFI = 1.000; TLI = 1.000; RMSEA = 0.000 (90% CI = 0.000–0.069), $p = 0.914$; and SRMR = 0.043.

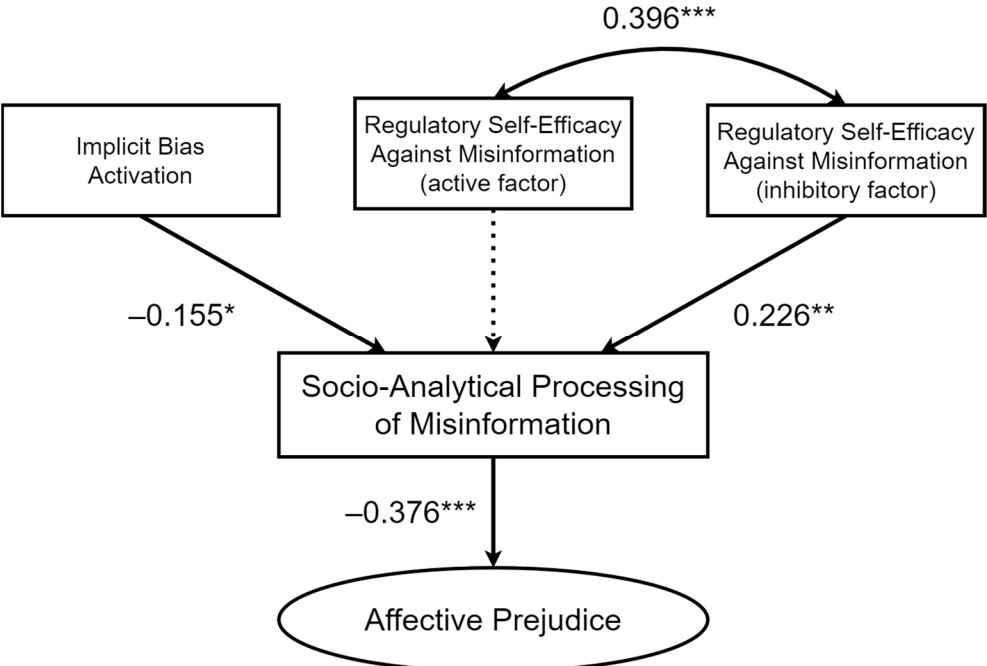

**Figure 4.** Structural Equation Model: robust maximum-likelihood standardized parameter estimates for the hypothesized model. Not shown in the diagram are the indicators of the affective prejudice construct, which were omitted for the purpose of visual clarity. * $p < 0.05$, ** $p < 0.01$, *** $p < 0.001$.

Regarding the measurement model, the findings show that the loadings between the latent stability factor of affective prejudice and the indicators (T1 and T2) obtained within a week of each other are 0.847 ($p < 0.001$) and 0.809 ($p < 0.001$), respectively. Considering the structural relationships between the variables under consideration, the results highlight that an increase in performance at the socio-analytical processing of misinformation is indeed significantly associated with a decrease in affective prejudice ($\beta = -0.376$, $p < 0.001$). Instead, no significant direct path was found between implicit bias activation and affective prejudice, and neither the inhibitory nor the active factor of regulatory self-efficacy was directly related to the main outcome. Conversely, the results show that as implicit bias activation increases, performance at socio-analytical processing decreases significantly ($\beta = -0.155$, $p = 0.033$). With regard to the two factors inhibitory and active self-efficacy, however, it can be observed that although they are significantly correlated with each other (0.396, $p < 0.001$), they exhibit substantially different relationships with respect to socio-analytical processing: while the link between performance at socio-analytical processing and the proactive factor of regulatory self-efficacy against misinformation does not reach the threshold of significance, the results show that as the inhibitory factor increases, processing performance also increases accordingly ($\beta = 0.226$, $p = 0.008$). To test for the presence of significant indirect effects of the inhibitory factor of regulatory self-efficacy and implicit bias activation on affective prejudice through socio-analytical processing, 95% confidence intervals for indirect effects were estimated by means of the bias-corrected bootstrap

method with 5000 samples (MacKinnon et al. 2004). The results, in this case, show the presence of a significant negative indirect effect of inhibitory regulatory self-efficacy on affective prejudice ($\beta = -0.085$, 95% CI [–0.157 –0.013]); in contrast, the confidence interval around the standardized estimate of the indirect effect of implicit bias activation on the main outcome via socio-analytical processing included zero ($\beta = 0.058$, 95% CI [–0.001 0.117]). As for the control variables, none of those considered significantly affected any endogenous variable in the model. Overall, the structural model explained 14.2% of the variance in affective prejudice.

## 4. Discussion

The results of the current study make multiple contributions to the existing literature on adolescent prejudice and misinformation. First, this study corroborates the findings of previous research (Lutzke et al. 2019; D'Errico et al. 2023), emphasizing the importance of analytical and reflective reasoning when confronting distorted online communication. Furthermore, it expands on these works by demonstrating the role self-regulative beliefs play in the socio-analytical processes that counteract biased cognitions. Secondly, we observed that the cognitive processes influencing socio-analytic reasoning confirm the dual processing pathway, even in the context of online disinformation promoting prejudice. It appears that while the automatic implicit activation of bias hinders socio-analytical reasoning, self-regulative beliefs pertaining to the management of online news facilitate it. Thirdly, we discovered that inhibitory self-efficacy, in particular, indirectly influences prejudice. This suggests that one's perceived ability to refrain from acting under certain circumstances could be pivotal in addressing biased information and the heuristic and intuitive processing it induces. In general, the present findings, which we will discuss in the subsequent paragraphs, illuminate how, during adolescence, self-directed and regulated analytical processes can be wielded to counteract automatic processes externally driven by distorted social content. This observation constitutes a novel practical contribution to education and prevention initiatives as it underscores the need to foster young individuals' confidence in their ability to critically analyze information and resist impulsively sharing news that reinforces the spread of stereotyped beliefs on social media.

### 4.1. Socio-Analytical Approach in Hindering Misleading Racial News in Adolescence

Online prejudice against migrants is an increasingly widespread issue, and past research has firmly established a correlation between the spread of misleading information on social networks and the rise of prejudiced behaviors (Ramasubramanian 2007). Building on preceding research [blinded] that identified socio-analytical interventions as effective in countering ethnic distorted cognitions linked to misinformation, we have once again underscored the significant role socio-analytical thinking plays in relation to affective prejudice among adolescents. Notably, this type of socio-analytical reasoning accentuates the importance of recognizing stereotyped beliefs and other prejudices inherent in misleading news concerning groups of people, thereby enabling a contextualized reasoning process that accounts for the recognition of group categorizations in online news, which can trigger heuristic processing. In doing so, this study contributes to our understanding of the specific individual factors potentially affecting adolescents' socio-analytical processing of misinformation. Consequently, by enhancing our understanding of the underlying mechanisms facilitating information processing in the online environment, we are better equipped to develop targeted strategies to combat ethnic bias and decrease aversion towards outgroups encountered online.

### 4.2. Dual Pathway toward Socio-Analytical Processing of Misinformation

The present study explores two hypothetical pathways of influence on socio-analytical thinking, each reflecting distinct operational modes: one associated with automatic information processing, and the other intentional and self-controlled. Consequently, it was hypothesized that socio-analytical reasoning about misinformation would be hindered by

adolescents' automatic activation, while self-beliefs about their capacity to self-regulate misleading information would enhance this reasoning. The study's findings partially support the dual-pathway hypothesis. Specifically, the activation of implicit attitudes, marked by automaticity and directed at particular social groups without awareness or intentionality, showed a negative association with the performance involving analysis and empathic perspective-taking inherent to socio-analytical reasoning. As noted earlier, certain terminologies, imagery, or verbal biases within misleading news can act as priming stimuli, triggering implicit biases and intensifying their effects (Strabac and Listhaug 2008; Epifania et al. 2023). Even though implicit attitudes often operate beyond conscious monitoring, the current findings suggest that adolescents are not entirely subject to automatic responses, a result of prolonged exposure to stereotype patterns that reinforce associative heuristics between social groups and attributes. In fact, in addition to automatic processes, this study's hypotheses emphasize the crucial role voluntary and self-aware factors play in credulity and countering related biases, as is the case with regulatory self-efficacy against misinformation. However, regarding the two components of these regulatory self-efficacy beliefs, adolescents' belief in their active, voluntary capacity to investigate news veracity seems unrelated to their socio-analytical performance in processing misleading news. This could be because adolescents have limited experience in this area, and their self-evaluation lacks grounding in sufficient efficacious experiences (Lönnfjord and Hagquist 2018). However, a correlation between this active factor and the inhibitory factor emerges. Indeed, the adolescent's self-perceived ability to inhibit the impulse to share potentially false news emerges as a facilitating factor for enhancing the socio-analytical processing of misleading news. This aligns with earlier research on countering distorted cognitions and harmful behaviors, which underscores the critical role of perceived inhibitory self-efficacy beliefs in adolescents (Bandura 2001). Given the intensification of social interactions and growing peer influence during adolescence (Albert et al. 2013; Mitic et al. 2021), susceptibility to peer pressure often increases, often leading to conformity that could result in an uncritical acceptance of information (Hjerm et al. 2018). The inhibitory aspect of self-efficacy serves as a vital protective mechanism, enabling adolescents to resist peer influence that might otherwise lead to the passive adoption and spread of misinformation. Therefore, by fostering beliefs grounded in effective experiences of inhibiting misinformation and promoting analytical and forward-thinking processing of online misinformation, adolescents can potentially mitigate prejudiced emotional responses.

*4.3. Implications and Shortcomings*

Understanding these individual factors and conceptualizing them as separate information processing pathways can aid educators and policy makers in crafting effective strategies to mitigate the impact of misleading news on adolescents. In particular, these strategies could encourage a slow and deliberate engagement with news, facilitated by beliefs associated with inhibitory mechanisms, as demonstrated by this study's results. Such an approach supports the socio-analytical processing of harmful misinformation that targets individuals from different cultural backgrounds. In addition to considering the role of automatic credulity processing (Pennycook and Rand 2019), this strategy also incorporates deliberate and analytical processing, thereby strengthening individuals' capacity to counter prejudiced misinformation, and consequently mitigating its adverse effects. For instance, when dealing with racial hoaxes, it becomes possible to implement preventive *pre-bunking* interventions. Here, adolescents, as potential news consumers and future information professionals, are equipped with the tools to recognize media biases (Lutzke et al. 2019; Paul and Elder 2004; Basol et al. 2020; D'Errico et al. 2023) and understand their long-term effects.

While the study offers valuable insights for prejudice intervention, it is important to consider some limitations. The study does not factor in other significant elements related to misinformation (e.g., technological and contextual affordances) and social and individual factors (e.g., online peer behaviors, emotional reactions, and existing knowledge) that could

influence the proposed process. Lastly, although ethnicity did not emerge as a significant predictor of the socio-analytical processing of misinformation or affective prejudice in our findings, this might be due to the limited size of the non-Caucasian subgroup. The research was conducted with a local sample of adolescents, underscoring the necessity to replicate the study in varied social and cultural contexts with more extensive adolescent samples, while also considering longitudinal research.

## 5. Conclusions

The findings underscore the intricate dynamics of adolescent engagement with online misinformation. Specifically, the two distinct information processing pathways significantly influence the extent of the socio-analytical processing of misinformation, and empirical data suggest that an enhanced level of such processing is correlated with reduced prejudice. Implicit activations and self-efficacy beliefs both play a role in shaping how young individuals analyze and respond to misleading content on digital platforms. Given the pervasiveness of misinformation in the digital landscape, this research highlights why it is vital to investigate the mechanisms that either exacerbate or mitigate online prejudice. Furthermore, as adolescents navigate the complexities of modern social network interactions, understanding these pathways becomes paramount in formulating strategies to boost socio-analytical processing, thus reducing prejudice.

**Author Contributions:** Conceptualization, G.C., P.G.C., F.D. and M.P.; methodology, G.C., P.G.C., F.D. and M.P.; software, G.C. and P.G.C.; formal analysis, G.C.; investigation, P.G.C.; data curation, G.C. and P.G.C.; writing—original draft preparation, G.C., P.G.C., F.D. and M.P.; writing—review and editing, G.C. and P.G.C.; visualization, G.C.; supervision, F.D. and M.P.; project administration, F.D.; funding acquisition, F.D. All authors have read and agreed to the published version of the manuscript.

**Funding:** This work was entirely supported by the European project 'STERHEOTYPES—STudying European Racial Hoaxes and sterEOTYPES' recently founded by 'Challenge for Europe' call for Project, Compagnia San Paolo (CUP: B99C20000640007).

**Institutional Review Board Statement:** The study was conducted in accordance with the Declaration of Helsinki and approved by the Ethics Committee of the University [blinded].

**Informed Consent Statement:** Informed consent was obtained from all subjects involved in the study.

**Data Availability Statement:** The dataset supporting the results of this study is available upon motivated request to the corresponding author [blinded]. The data are not publicly available because, although completely anonymous, their public disclosure is not specifically mentioned in the informed consent provided by the participants on the use of confidential data.

**Conflicts of Interest:** The authors declare no conflict of interest.

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
