# Peer review of "Preventing Prejudice Emerging from Misleading News among Adolescents: The Role of Implicit Activation and Regulatory Self-Efficacy in Dealing with Online Misinformation"

_socsci, doi:10.3390/socsci12090470_

Round 1

Reviewer 1 Report

This paper has an interesting and important focus, but suffers from structural issues that get in the way of its contribution. For example, the introduction seems to be less an introduction and more a combined introduction and literature review. The result feels a bit cluttered and muddied. I wish the authors would begin this with a more formal introduction -- a three to four paragraph description of what this study seeks to examine, what it finds, and why those findings matter. From there, I'd hope to see a literature review with distinct sections exploring each of the components that the authors are most interested in: information processing, misleading news, and prejudice. Once the authors have unpacked each of these factors individually, then the literature review might end with a final section that answers: How do each of these things come together?

I'd like to see a similar approach to revising the discussion section, which also feels cluttered and disorganized. It begins with just a massive paragraph that is in need of being broken up into smaller ones. Again, the authors might consider using section subheads to help guide the reader. The topics the authors are focused on here are complex and difficult to fully unpack, and it's even harder for the reader to really get where the authors are coming from if each of these topics feels rolled into the next rather than given the space and attention to breathe. 

Finally, I'd like the authors to spend a bit more space dedicated to describing the implications of their findings. How do their results help ongoing discussions within media, journalism, and audience studies when it comes to questions surrounding the accuracy of news media reported by journalists, the interpretation of news media by news audiences, and role of prejudice in both sides of that equation?

All in all, I think this project's goals are worthwhile, but more needs to be done to make the contribution clearer.

Author Response

Dear Reviewer, thank you for your constructive feedback on our manuscript.

This paper has an interesting and important focus, but suffers from structural issues that get in the way of its contribution. For example, the introduction seems to be less an introduction and more a combined introduction and literature review. The result feels a bit cluttered and muddied. I wish the authors would begin this with a more formal introduction -- a three to four paragraph description of what this study seeks to examine, what it finds, and why those findings matter. From there, I'd hope to see a literature review with distinct sections exploring each of the components that the authors are most interested in: information processing, misleading news, and prejudice. Once the authors have unpacked each of these factors individually, then the literature review might end with a final section that answers: How do each of these things come together?

In response to the first point, we took your advice and significantly restructured the introduction after a constructive discussion among the authors. It now provides a broader overview of the theoretical and methodological background of the paper, with separate sections exploring the overall aim of the paper, then 
addressing prejudice and misleading news, then information processing. We hope that this improved layout will help make our introduction more coherent and clear.

I'd like to see a similar approach to revising the discussion section, which also feels cluttered and disorganized. It begins with just a massive paragraph that is in need of being broken up into smaller ones. Again, the authors might consider using section subheads to help guide the reader. The topics the authors are focused on here are complex and difficult to fully unpack, and it's even harder for the reader to really get where the authors are coming from if each of these topics feels rolled into the next rather than given the space and attention to breathe. 

Regarding your comments on the discussion section, we have taken your advice into consideration. We have similarly broken the larger paragraphs into smaller sections with clear subheadings. Additionally, we improved the transitions between concepts and decluttered the language for the sake of clarity and readability.

Finally, I'd like the authors to spend a bit more space dedicated to describing the implications of their findings. How do their results help ongoing discussions within media, journalism, and audience studies when it comes to questions surrounding the accuracy of news media reported by journalists, the interpretation of news media by news audiences, and role of prejudice in both sides of that equation?

Your suggestion to elaborate on the implications of our findings was well-received: we expanded on this aspect in the discussion section, particularly under the sub-section "Implications and Shortcomings." Here, we emphasize the relevance of our study to the ongoing discussions, particularly its potential to inform interventions that empower adolescents as active agents against misinformation, fostering their development as future citizens and information professionals.

All in all, I think this project's goals are worthwhile, but more needs to be done to make the contribution clearer.

We sincerely thank you for recognizing the merit of our research goals, and for your valuable guidance in enhancing the clarity of our paper. We believe that your comments have significantly improved our manuscript and have made our contributions more explicit.

Reviewer 2 Report

Dear Author/Authors, 

This journal article covers a pertinent topic that must be studied. The article contains significant level of technical description of the study that you carried out with school students. In fact, it is this fact that would make the study's findings and analysis inaccessible to many. It is too technical. My suggestion would be to modify the overall language used in the article (if possible shorten some of the long sentences for clarity) and simplify the description of various technical explanation of the methodology used for this study.

Also needed are the details of how students from different schools and different demographic backgrounds performed in your study. Was the age, socio-economic background or individual student's own racial background impacted on their response to misinformation about certain race/s? In relation to the second measure, which group of students modified their prejudiced view of other race/s more than other group of students?   

The article is too technical.

Author Response

Dear Author/Authors, 

This journal article covers a pertinent topic that must be studied. The article contains significant level of technical description of the study that you carried out with school students. In fact, it is this fact that would make the study's findings and analysis inaccessible to many. It is too technical. My suggestion would be to modify the overall language used in the article (if possible shorten some of the long sentences for clarity) and simplify the description of various technical explanation of the methodology used for this study.

Dear Reviewer, thank you for your constructive comments and suggestions.

We've taken into account your feedback regarding the readability of our paper. In response, we have undertaken a thorough revision to modify the language and structure of the paper. Our aim was to make the discussion and introduction less technical, while preserving the necessary technical aspects for the statistical analysis in the methodology section. We hope these adjustments enhance clarity for a diverse readership. Additionally, we simplified the sentence structures, clarified the underlying logical steps, and reduced overall complexity. 

Also needed are the details of how students from different schools and different demographic backgrounds performed in your study. Was the age, socio-economic background or individual student's own racial background impacted on their response to misinformation about certain race/s? In relation to the second measure, which group of students modified their prejudiced view of other race/s more than other group of students?   

Thank you for your observations regarding the potential effects of demographic variables. Your inquiries prompted us to present a more comprehensive description of our adolescent sample and to further refine our analysis. First of all, we included data on students' ethnic background and parental educational qualifications in the “Participants” paragraph, which we believe provides a richer understanding of our adolescent sample. Then, we adjusted our model to account for the impact of students' ethnic backgrounds: our analyses show that the significant pathways remain unaltered, and the ethnic background itself does not explain a statistically significant amount of variance in socio-analytical performance or affective prejudice. Lastly, similar analyses were carried out considering parental educational qualifications, and they also showed no statistically significant impact. 

We do believe that these revisions have substantially improved our manuscript, making it both more clear and comprehensive. Thank you once again for your constructive feedback.